# Cytotoxicity and Genotoxicity of Metal Oxide Nanoparticles in Human Pluripotent Stem Cell-Derived Fibroblasts

**Harish K Handral** [1,2], **C. Ashajyothi** [3], **Gopu Sriram** [1], **Chandrakanth R. Kelmani** [4], **Nileshkumar Dubey** [1,5,*] and **Tong Cao** [1,*]

1    Faculty of Dentistry, National University of Singapore, Singapore 119085, Singapore; mpehkh@nus.edu.sg (H.K.H.); dengs@nus.edu.sg (G.S.)
2    Department of Mechanical Engineering, National University of Singapore, Singapore 117575, Singapore
3    Department of Biotechnology, Vijayanagara Sri Krishnadevaraya University, Ballari, Karnataka 583105, India; drashajyothic@vskub.ac.in
4    Department of Biotechnology, Gulbarga University, Gulbarga, Karnataka 585106, India; ckelmani@gmail.com
5    Department of Cariology, Restorative Sciences and Endodontics, School of Dentistry, University of Michigan, Ann Arbor, MI 48104, USA
*    Correspondence: nileshd@umich.edu (N.D.); dencaot@nus.edu.sg (T.C.)

**Abstract:** Advances in the use of nanoparticles (NPs) has created promising progress in biotechnology and consumer-care based industry. This has created an increasing need for testing their safety and toxicity profiles. Hence, efforts to understand the cellular responses towards nanomaterials are needed. However, current methods using animal and cancer-derived cell lines raise questions on physiological relevance. In this aspect, in the current study, we investigated the use of pluripotent human embryonic stem cell- (hESCs) derived fibroblasts (hESC-Fib) as a closer representative of the in vivo response as well as to encourage the 3Rs (replacement, reduction and refinement) concept for evaluating the cytotoxic and genotoxic effects of zinc oxide (ZnO), titanium dioxide (TiO$_2$) and silicon-dioxide (SiO$_2$) NPs. Cytotoxicity assays demonstrated that the adverse effects of respective NPs were observed in hESC-Fib beyond concentrations of 200 µg/mL (SiO$_2$ NPs), 30 µg/mL (TiO$_2$ NPs) and 20 µg/mL (ZnO NPs). Flow cytometry results correlated with increased apoptosis upon increase in NP concentration. Subsequently, scratch wound assays showed ZnO (10 µg/mL) and TiO$_2$ (20 µg/mL) NPs inhibit the rate of wound coverage. DNA damage assays confirmed TiO$_2$ and ZnO NPs are genotoxic. In summary, hESC-Fib could be used as an alternative platform to understand toxicity profiles of metal oxide NPs.

**Keywords:** human pluripotent stem cells; fibroblasts; nanoparticles; cytotoxicity; genotoxicity

## 1. Introduction

Advances in nanotechnology are proceeding at a striking rate and a diverse number of industrial, commercial, consumer, and health-care products have expanded by incorporating nanomaterials into these products [1,2]. Due to the large number, diversity, and use of NPs (nanoparticles), toxicology studies cannot keep pace. Since most of the past and current models largely depend on mammalian animals for testing, such time- and resource-intensive studies, albeit informative, cannot assess the avalanche of NP-enabled technology. Owing to differences in the toxicity profiles of the native and nano-sized counterparts, risk-assessment frameworks specific to NPs that incorporate alternative testing models have been proposed [3]. Secondly, reduction and replacement of animal use has been promoted to adopt 3Rs in (Organisation for Economic Co-operation and Development) OECD guidelines and has led to calls for tiered and integrated test strategies that were used to screen and assess NPs along their chemical life cycle by in silico, in vitro and other alternative models [4].

Due to high volumes of testing and limited availability of primary human cells, cell lines of animal-origin, immortalized, and/or cancer-derived sources are commonly used

for high-throughput toxicology screening studies. Since cellular immortality leads to epigenetic changes, tumorigenesis, chromosomal and genetic aberrations, the physiological relevance and reliability of cancer-derived and immortalized cell lines is limited [5,6]. Further, though primary human cells are a good platform to rely on procuring large amounts of primary cells typically needed for high-throughput assays is challenging. On the other hand, pluripotent stem cells such as human embryonic stem cells (hESCs), due to their long-term self-renewal and pluripotent ability to derive differentiated lineages, are a potential source for obtaining unlimited amounts of somatic cells [3]. Additionally, hESCs have been shown to be genetically and karyotypically normal, thus considering hESCs as a representative source of how the normal cell acts in vivo. There are some similar reports toxicity of various ZnO, $TiO_2$, and $SiO_2$ NPs. The comparative toxic effect of this nanoparticle of hESC-derived fibroblast is nascent and yet to be explored and requires further investigation [7]. Thus, in this study, we explored the potential use of human embryonic stem cell-derived fibroblasts (hESC-Fibs) to evaluate the cellular toxicology profiles of most common metal oxide NPs.

## 2. Materials and Methods

### 2.1. Nanoparticle Stock Preparation and Characterization

Nanoparticles $TiO_2$ (product number: 791326, <100 nm), $SiO_2$ (product number: 637238, 10–20 nm), and ZnO (product number: 721077, <100 nm) of sizes were purchased from Sigma Aldrich (St. Louis, MO, USA). A stock dispersion of as received nanoparticles was weighed on an analytical mass balance, suspended in deionized water at a concentration of 50 mg/mL, and then probe sonicated for 2 min at 35–40 W to assist in mixing and forming a homogeneous dispersion. For further use, the stock concentration was diluted to prepare working concentrations in DMEM-high glucose media with 10% FBS and 1% pen/strep. NPs were spread on a silicon wafer to characterize using Scanning Electron Microscopy (SEM) (Verios XHR SEM, FEI, Waltham, MA, USA) and Transmission Electron Microscopy (TEM) (Tecnai 20 G2, CSIR-CECRI, Karaikudi, India).

### 2.2. Cell Culture and Differentiation

H1 hESCs were purchased from WiCell Research Institute, (Madison, WI, USA) and cultured using mTESR1$^{TM}$ culture media (Stem Cell Technologies$^{TM}$, Vancouver, BC, Canada) by using Matrigel$^{TM}$ coated 6-well tissue culture plates as previously described [8–10]. Our group's established protocol was adapted to obtain hESC-Fib [9–13]. Briefly, hESC colonies were dissociated using dispase (1 mg/mL, Stem Cell Technologies) and plated on ultra-low attachment plates to form embryoid bodies, during which low-glucose DMEM (Biowest, Riverside, MO, USA) culture media was fed for 10 days. On day 11, floating embryoid bodies were seeded on 0.1% gelatin-coated plates, and cells were nourished with DMEM-high glucose (Biowest, Riverside, MO, USA) culture media supplemented with 10% fetal bovine serum (FBS) (Biowest, Riverside, MO, USA). Cells were cultured for 2 weeks until the fibroblast-like outgrowths have emerged, after which, cells were passaged and cultured in high-glucose medium for two more passages before using them for experiments.

Characterization studies for hESC-Fib in comparison with commercially available human primary fibroblasts (PromoCell, Heidelberg, Germany) for morphological features, RT-PCR-based gene expression as well as protein level expression by immunofluorescence were performed.

### 2.3. Assessment of Cellular Viability

The effect of NPs on viability of cells was accessed by MTS assay (CellTiter 96$^{®}$ AQueous One Solution Cell Proliferation Assay, Promega, Madison, WI, USA) and trypan blue dye exclusion method. In MTS assay, cells were cultured in 96-well plates (1000 cells in 100 μL media/well) until they reach 80% confluency and cells were treated with desired concentrations of metal oxide NPs. After 24 h of NP treatment, 20 μL of MTS reagent in 100 μL of media was added to each group and incubated for 4 h. Culture medium without

NPs was considered as negative control. DMSO was used as the reference toxicant and considered as positive control. Absorbance was read at 490 nm using plate reader (M200 Infinite, Tecan, Switzerland). Cell viability was represented as percentage of viable cells with respect to no treatment control.

For trypan blue assay, cells were treated with respective concentrations of metal oxide NPs for 24 h and stained with trypan blue dye. The number of live and dead cells were manually counted using hemocytometer and cellular viability was expressed as percentage of viable cells.

### 2.4. Dose-Response Curves

From the MTS assay results, dose-response curves were plotted to investigate no observable adverse effect concentration (NOAEC), inhibitory concentration ($IC_{50}$) and total lethal concentration (TLC) values [14]. To calculate these values, normalized cell viability (viability of cells in normal culture medium = 1) was plotted as a function of logarithm of the particle. Theoretical dose-response curves were extrapolated using the following equation:

$$y = 1 - 1/1 - e^{a(b-x)} \tag{1}$$

where, $a$ = curve slope, $b$ = $IC_{50}$ value (in μg/mL) and $x$ is the absorbance values expressed in percentage of cell viability.

NOEAC values were determined considering $y$ value as 95%, maximum cell viability in normal cell culture medium. Equation is as follows:

$$\text{NOEAC} = b - \ln(19)/a \tag{2}$$

Similarly, TLC value was determined with $y$ being 5% of the maximum response on the normalized viability. Equation is as follows:

$$\text{TLC} = b - \ln(1/19)/a \tag{3}$$

### 2.5. Assessment of Membrane Potential Using Lactate Dehydrogenase Assay (LDH)

CytoTox 96® Non-Radioactive Cytotoxicity Assay kit (Promega, Madison, WI, USA) was used to access the membrane integrity of cells by measuring the release of LDH. Briefly, cells cultured in 96-well plates were treated with respective concentrations of NPs for 24 h. The 100 μL of cell-free supernatant were subsequently transferred from each well into a 96-well plate, and the LDH reaction mixture was added to each well in triplicates. After incubating for 3 h, absorbance values of supernatant were obtained at 490 nm wavelength using plate reader (M200 Infinite, Tecan, Seestrasse, Männedorf, Switzerland).

### 2.6. Detection of Apoptotic Cells

Flow cytometry analysis was used to detect apoptotic/necrotic profile of NP-treated cells. Double staining of APC Annexin V/Propidium iodide (PI) was done by following manufacturer's instructions of Annexin-PI kit (Invitrogen, ThermoFisher, Waltham, MA, USA). Briefly, hESC-Fib cells treated with respective concentrations of metal oxide NPs were trypsinized after 24 h of NPs' treatment and cell pellet was stained with 4 μL of AnnexinV and incubated in dark at 4 °C for 15 min. Later, cells were washed with 200 μL of FACS buffer and centrifuged to remove an unbound stain; after which, cells were suspended in 500 μL of FACS buffer and stained with 5 μL of PI (1 mg/mL). Stained cells were taken for flow cytometry analysis (CyAn Analyzer, Brea, CA, USA). Based on previously published reports, percentage of apoptotic cells was calculated [15,16].

### 2.7. Assessment of Cellular Proliferation Using Scratch Wound Assay

Scratch wound assay was performed to assess the ability of metal oxide NPs to promote or inhibit cell proliferation. The hESC-Fib were cultured as confluent monolayers in 12-well plates and wound scratch was made with sterile 100 μL-micropipette tips.

Culture plates were rinsed with PBS before treating the cells with NPs. Cells with no NP treatment were considered as control. After 24 h of NPs' treatment, mean distance between wound edges was measured by imaging under an inverted phase contrast microscope (Olympus IX70 fluorescence microscope, Shinjuku City, Tokyo, Japan). Images were analyzed by using commercially available image analysis software (TScratch software, developed by the group of Dr. Koumoutsakos (CSE Lab), at the Eidgenössische Technische Hochschule (ETH), Zurich, Switzerland [17].

### 2.8. DNA Damage Assay for Genotoxic Profile

HCS DNA damage assay kit (Invitrogen Carlsbad, CA, USA) was used to analyze genotoxic nature of NPs on hESC-Fib. The hESC-Fib were cultured till 80% confluency. Cells were treated with respective metal oxide NPs for 24 h. We have considered mild and $IC_{50}$ concentrations of all three NPs. After NP treatment, cells were fixed with 4% paraformaldehyde and stained using HCS DNA damage kit as per manufacturer's instructions. Mechanism of the staining works was by phosphorylated $\gamma$H2AX which is conjugated with Alexa fluor 555; foci formed at the damaged DNA site in the nucleus is measured by specific antibody-based detection and Hoechst 33,342 (a DNA binding dye) stains nucleus to highlight the changes in nuclear morphology of healthy as well as damaged cells.

## 3. Statistical Analysis

For the cytotoxicity evaluation, concentration response curves were fitted. Statistics were performed with One-way ANOVA and Tukey test ($\alpha$ = 0.05).

## 4. Results

### 4.1. Characterization of Nanoparticles (NPs)

SEM and TEM analysis demonstrated spherical shape of $SiO_2$ and $TiO_2$ NPs with size ranging from 10–20 nm and <100 nm, respectively; with marginal variation particles showed poly-aggregation. Electron microscopy analysis report confirmed the features of ZnO NPs (size < 100 nm) with rod-shaped morphology. Figures 1 and 2, highlight the dimension and surface morphology of the nanoparticle samples examined by SEM and TEM.

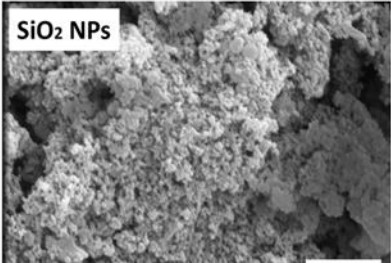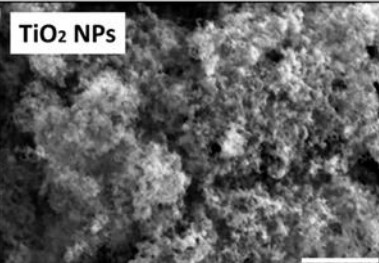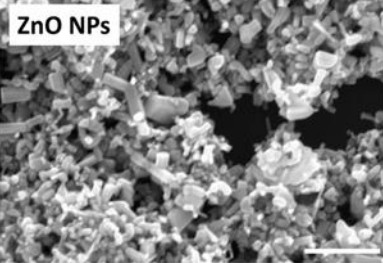

**Figure 1.** SEM (scanning electron microscopy) photomicrographs of $SiO_2$ NPs (nanoparticles) deposited on the substrate formed the agglomerates consisting of the tangled chains with the particle size ranging from 10–20 nm, spherical shape of $TiO_2$ NPs with an average crystallite size of about 20 nm and ZnO NPs with an average size of 100 nm. Scale bar: 500 μm.

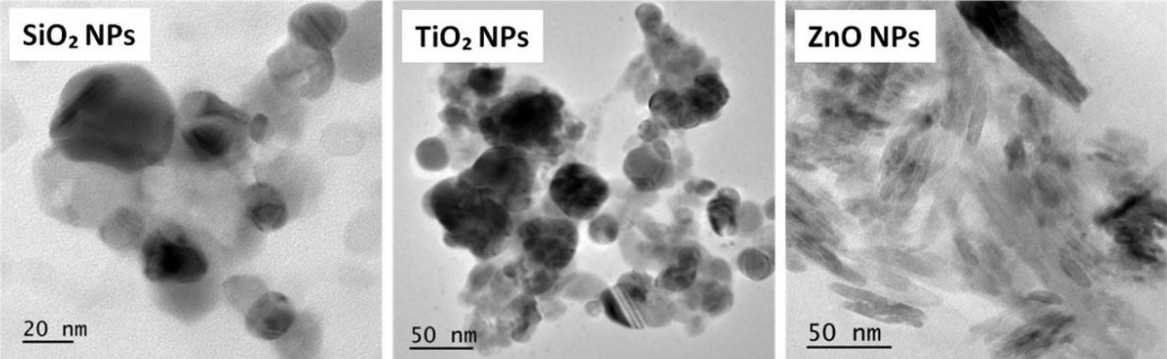

**Figure 2.** TEM photomicrographs of SiO$_2$, TiO$_2$ and ZnO NPs. Highlighting the spherical shape in case of SiO$_2$ and TiO$_2$ NPs. ZnO NPs showing rod-shaped structures.

### 4.2. Characterization of hESC-Fib

The hESC-Fib cells were characterized for morphological features in comparison with primary fibroblasts (Figure 3a). Vimentin expression was confirmed by immunofluorescence staining (Figure 3b) and RT-PCR studies (Figure 3c).

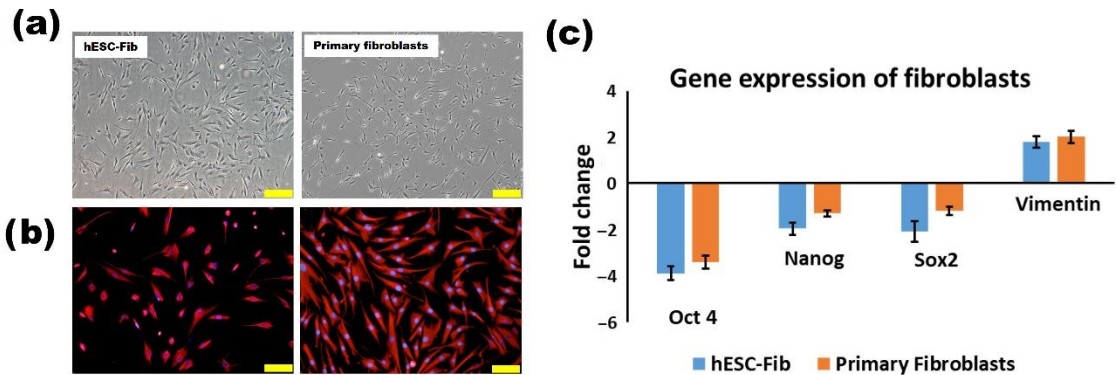

**Figure 3.** Characterization analysis of hESC-Fib (human embryonic stem cell-derived fibroblast): (**a**) Phase contrast microscopic image of hESC-Fib and primary fibroblasts cultured on gelatin-coated tissue culture plate. Phase contrast images show the spindle-shaped morphology of cells; (**b**) Immunofluorescence-stained photomicrographs of hESC-Fib and primary fibroblasts expressing the presence of Vimentin in red and DAPI stained nucleus in blue; (**c**) Real-time PCR of hESC-Fib compared to primary fibroblasts and both cells were also checked for expression of pluripotent genes. Levels of expression were normalized by corresponding GAPDH values, an internal control. Scale bar: 200 μm.

### 4.3. Cellular Viability in Presence of NPs

The SiO$_2$, TiO$_2$ and ZnO NP-treated cells displayed morphology close to that of untreated fibroblasts at a concentration of 50, 10, and 5 μg/mL. However, detrimental effect on cell morphology and even cell death was observed with higher NP concentrations (i.e., 1250, 250 and 50 μg/mL SiO$_2$, TiO$_2$ and ZnO NPs respectively). Phase contrast images showed that higher concentrations of SiO$_2$ and TiO$_2$ NPs have disrupted the cell's morphology, whereas ZnO NPs have led to cell death. This implies SiO$_2$ and TiO$_2$ NPs could be cytotoxic at higher concentrations, however, ZnO NPs have shown cytotoxicity at lower concentrations (Figure 4).

As per trypan blue dye exclusion assay, ZnO NPs at and above 30 μg/mL showed sudden decrease in cell viability. In the case of cells treated with TiO$_2$ and SiO$_2$ NPs, there has been steady decline in cell viability with a significant increase in NP concentrations (Figure 5). Similarly, MTS and LDH assay results co-relate with results of trypan blue assays. Cells treated with SiO$_2$ and TiO$_2$ NPs show less LDH release and percentage of toxicity at lower concentrations, however, ZnO NPs at lower concentration (5 μg/mL) have

shown 40% cellular toxicity. LDH assay provided a better view of cytotoxic nature of NPs affecting cell membrane integrity (Figure 5). In addition, the optimal dose level for each NP was also calculated (Table 1).

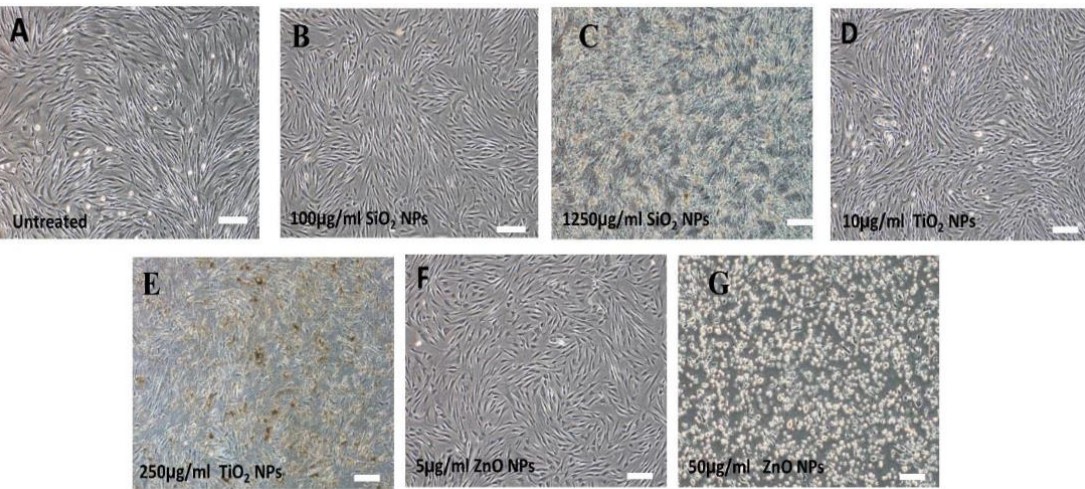

**Figure 4.** Phase contrast images of hESC-Fib cells taken after 24 h treatment of NPs, where (**A**) was considered as control i.e., cells with no NP treatment (untreated group). This figure also highlights the cell disruptions upon treatment of lower and higher concentrations of $SiO_2$ (**B,C**), $TiO_2$ (**D,E**) and ZnO (**F,G**) NPs. Scale bar: 100 μm.

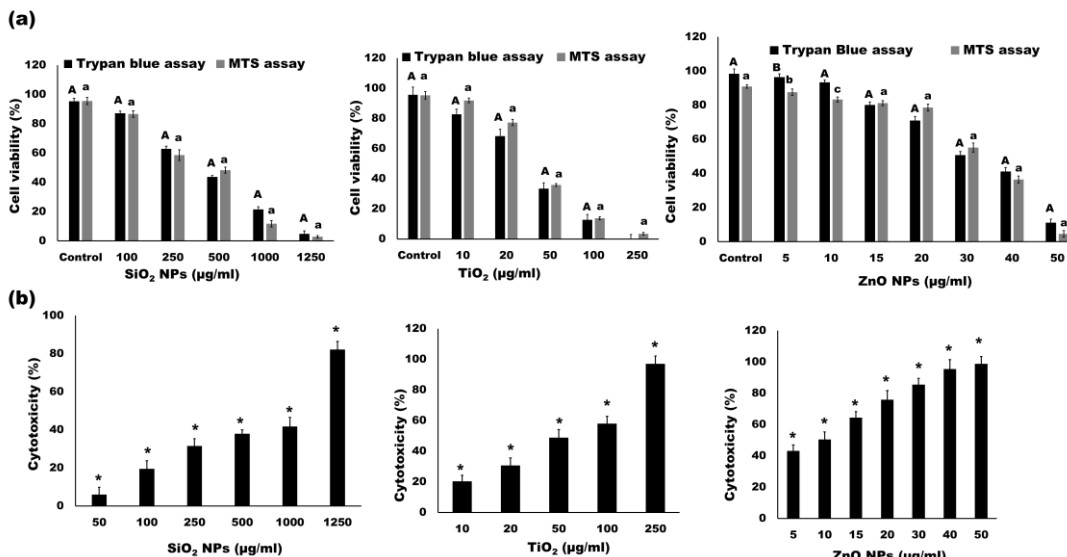

**Figure 5.** Represents the cytotoxic profiles of hESC-Fib in presence of NPs: (**a**) Trypan blue assay and MTS assay highlights the cell survival. The same letters indicate a significant difference compared with the control. "A" denotes significance at $p < 0.05$ for Trypan blue assay and "a" denotes significance at $p < 0.05$ for MTS assay; (**b**) LDH assay depicts the membrane potential of hESC-Fib cells after NP treatment. (* represents statistical significance compared to control).

**Table 1.** NOAEC (no observable adverse effect concentration), $IC_{50}$ (inhibitory concentration), and TLC (total lethal concentration) values for hESC-Fib cells cultivated with $SiO_2$, $TiO_2$ and ZnO NPs.

| NPs | NOEAC | $IC_{50}$ | TLC |
|---|---|---|---|
| $TiO_2$ (μg/mL) | 49.705 ± 2.4 | 50 ± 3.1 | 52.39 ± 4 |
| ZnO (μg/mL) | 39.412 ± 2 | 40 ± 3.2 | 42.39 ± 1.3 |
| $SiO_2$ (μg/mL) | 245.98 ± 6 | 250 ± 4.3 | 255.39 ± 7 |

### 4.4. Apoptosis Assay

The percentage of cells undergoing apoptosis/necrosis was investigated using flow cytometric analysis (Figure 6). Annexin-V and Propidium Iodide staining kit (Alexa Fluor®, ThermoFisher Scientific, Waltham, MA, USA) was used to analyze the cell apoptotic profile. Cells treated with ZnO NPs have shown apoptosis at 5 μg/mL. Approximately 20%–30% of cells were viable under 20 μg/mL concentration group, whereas cells treated with $TiO_2$ and $SiO_2$ NPs have shown lesser viability only in higher concentrations, i.e., 100 μg/mL and 500 μg/mL respectively. Figure 6 depicts the gradual increase in apoptotic cell population with gradual increase in NP concentration. Flow cytometry analysis graphs are mentioned in Supplementary Materials Figure S1. All the tested NPs at higher concentrations have induced apoptosis in hESC-Fib cells ($p \leq 0.05$ and $p \leq 0.001$). Thus, ZnO NP-treated cells showed strong apoptotic response as compared to $TiO_2$ and $SiO_2$ NPs.

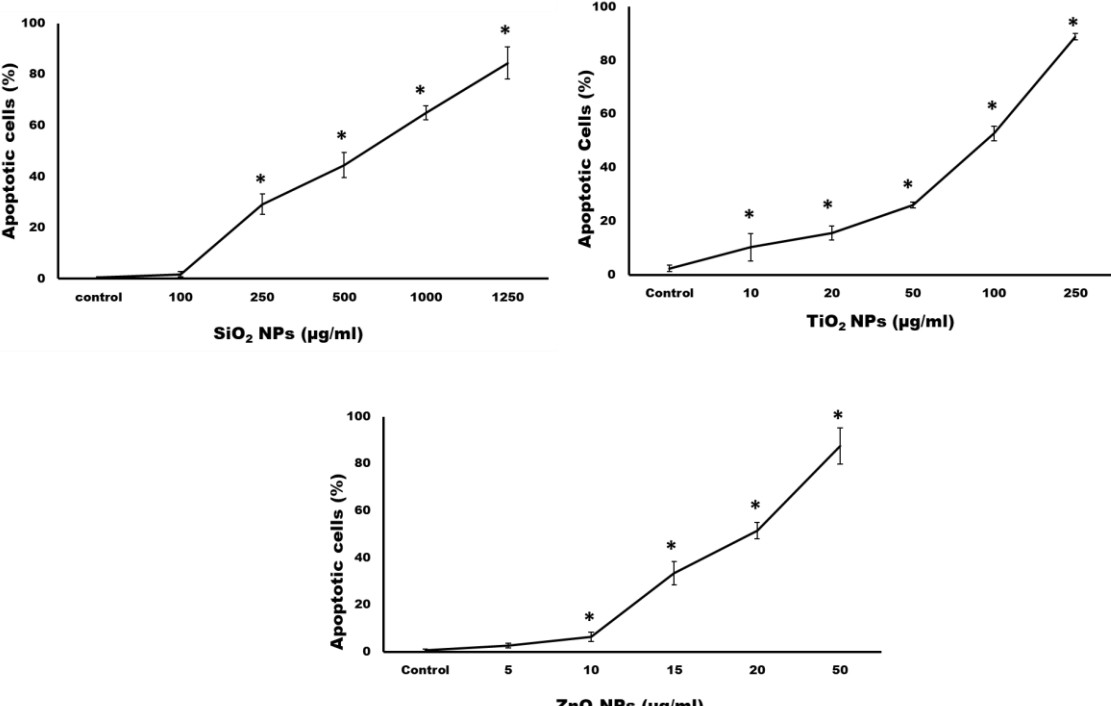

**Figure 6.** Effect of nanoparticles on the apoptosis of hESC-derived fibroblasts exposed to different concentrations in culture after 24 h. (* represents statistical significance compared to control).

### 4.5. Wound Healing Scratch Assay

Scratch assay was performed to address the impact of metal oxide nanoparticles on wound healing and collective migratory behavior of hESC-Fib cells. From Figure 7, $TiO_2$ and ZnO NPs at 20 μg/mL and 15 μg/mL showed approximately 50% of wound closure, respectively. Surprisingly, $SiO_2$ NPs did not show any cell proliferation beyond 250 μg/mL. The percentage of the wound covered over a period was calculated using TScratch software [17].

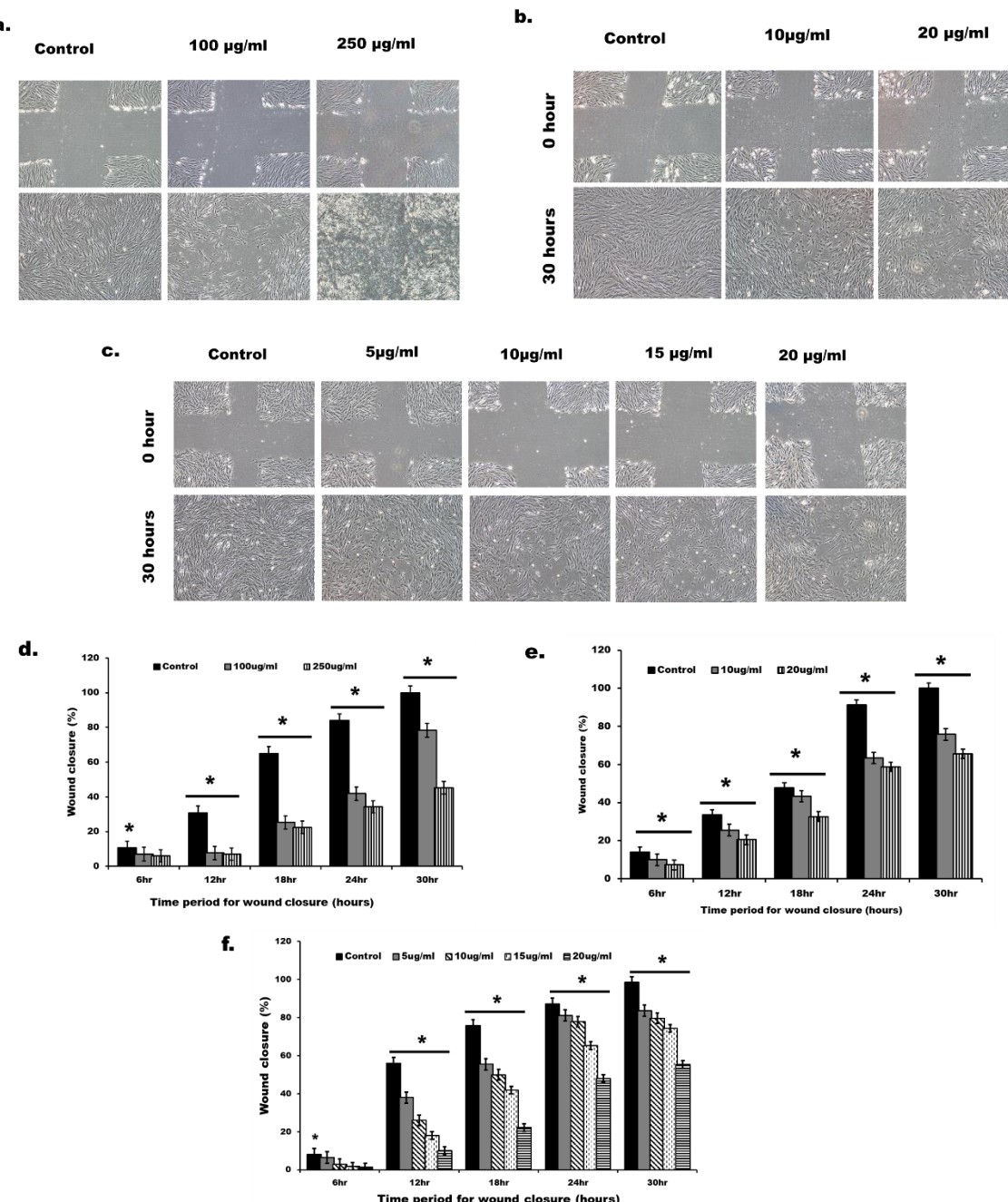

**Figure 7.** Wound healing scratch assay showing the cell proliferation and migration capability when treated with $SiO_2$ NPs (**a**,**d**); $TiO_2$ NPs (**b**,**e**) and ZnO NPs (**c**,**f**). (* represents statistical significance compared to control).

### 4.6. Genotoxicity Assay

Given the cell viability results in Figures 4 and 5 and dose response concentrations in Table 1, we investigated genotoxic effects of metal oxide NPs on hESC-Fib at NOEAC and $IC_{50}$ concentrations. Figure 8 shows the fluorescent images of nuclear/DNA damages in the NP-treated hESC-Fib cells. Subsequent increase in NP concentration changed the nuclear morphology, which is distinctly visible in ZnO NP-treated cells at 20 μg/mL as compared to $TiO_2$- (50 μg/mL) and $SiO_2$- (250 μg/mL) treated cells. ZnO and $TiO_2$ NPs at $IC_{50}$ concentration have shown pH2AX (TRITC/Alexa Fluor 555) signals, but only ZnO NP-treated hESC-Fib cells showed a shrunken nucleus. In summary, HCS DNA damage assay highlights the genotoxic response of ZnO NPs at $IC_{50}$ concentration. However, $TiO_2$ and $SiO_2$ NPs have not shown any major change.

## 5. Discussion

For evaluating NP-induced mechanism, cell-based in vitro toxicity testing is a very important alternative testing tool that could reduce animal model experimentation in most of the toxicology studies [5]. However, the responses to NPs are known to be dependable on cell type, cell's physiological action and NP size, because diverse physicochemical properties of NPs are known to trigger certain cellular changes [18]. This study aimed to use hESC-derived cells as an alternative cell source to evaluate the toxic effects of commonly used metal oxide NPs. NPs were dispersed in serum containing cell culture media and used to check the effects of NPs on cells. We differentiated hESCs to fibroblasts under in vitro conditions. NP toxicity was assessed by MTS, trypan blue dye exclusion assay and LDH assay for cytotoxic response.

The $SiO_2$, $TiO_2$, and ZnO NPs are known for their salient features such as porosity, specific surface characteristics, surface functionalization property and biocompatibility [19]; hence are widely used for biomedical applications, especially in drug delivery, gene delivery, nucleic acid extraction and imaging [20]. From previous reports, silica NPs show toxic effects on human and mouse cells at concentrations beyond 100 μg/mL after 24 h of exposure by indicating the cell lysis after 24 h of incubation, and also cellular agglomeration was observed [21,22]. Similarly, our studies revealed hESC-Fib cells treated with $SiO_2$ NPs for 24 h showed approximately 80% of viability at 100 μg/mL (Figure 5).

ZnO and $TiO_2$ NPs are more toxic than $SiO_2$ NPs. ZnO NPs were reported to initiate cytotoxicity from the range of 5–100 μg/mL in epithelial cells [23]. The rod-shaped ZnO nanoparticles have shown to have higher toxicity compared to the corresponding spherical ones [24]. Thus, nano-rod ZnO particles were used to assess the toxicity response on hESC-derived fibroblasts compared to spherical-shaped $SiO_2$ or $TiO_2$ to induce significant toxicity. In the current study, hESC-Fib cells treated with ZnO and $TiO_2$ NPs showed similar cytotoxic effects at the approximately same concentrations. Likewise, small-sized $SiO_2$ nanoparticles are widely used compared to bigger size and have been proved more toxic than bigger $SiO_2$ nanoparticles [25]. Cell viability results were confirmed by trypan blue assay, MTS assay as well as LDH assay (Figure 5). Trypan blue dye exclusion assay, the simplest assay to determine percentage of cell viability and assay, emphasizes the live cells having an intact cell membrane which excludes dyes/stains to enter, whereas dead cells take up the stain [26]. Similarly, MTS assay measures cell viability based on biological reduction of MTS tetrazolium compound and conversion into a formazan product. Amount of formazan produced is directly proportional to the number of live cells, which was measured by colorimetric analysis using a plate reader [27]. MTS assay showed approximately 50% of viability in the cells treated with 250 μg/mL of $SiO_2$ NPs. Followed by MTS assay, dose response concentrations were calculated for each nanoparticle (Table 1). Previously, the effects of NPs were evaluated also by the release of enzyme lactate dehydrogenase (LDH). LDH, a stable cytosolic enzyme that releases upon cell lysis. LDH assay represented the damages in cell membrane induced by NPs [28]. In our study, LDH assay confirms the toxic concentrations of metal oxide NPs to hESC-Fib cells. ZnO NPs are shown to be more toxic among the three NPs at lower concentration (15 μg/mL) as compared to $TiO_2$ and $SiO_2$ NPs. Cytotoxicity assays also explain the speed of toxic response by ZnO NPs. ZnO NPs at 10 μg/mL has damaged nearly 50% of cells in 24 h of exposure, whereas $TiO_2$ NPs are five times slower and $SiO_2$ NPs are 50 times. $TiO_2$ NPs are one of the most commonly used NPs in biomedical industry either in cosmetics or as Ti implants. Interestingly, only 100 μg/mL of $TiO_2$ NPs upon 24 h of exposure have shown 75–80% of cytotoxic effects in hESC-Fib cells. Based on the cytotoxic response, $IC_{50}$ and TLC values of NPs were calculated. $IC_{50}$ concentrations of NPs were used in further analysis. Various physical factors of NPs such as shapes and geometries including spheres, ellipsoids, cylinders, sheets, cubes, spikes and rods considerably induce toxicity to the cells. In relation to this, we have shown that rod-shaped ZnO NPs at 50 μg/mL have more capacity to induce physical damage to cells leading to higher necrosis than spherical-shaped $TiO_2$ and $SiO_2$ NPs at 50 μg/mL. Likewise, in cases of size-dependent nanotoxicity studies, it

has been known that distribution of NPs among cells/tissues in an in vivo scenario, that significant difference in distribution of NPs has been reported. Lower-size NPs have been found at most cellular sites, but higher-size NPs at limited locations are one of the main reasons why small-size NPs have been chosen for drug delivery applications. Considering this, toxicity evaluation of small-sized NPs is important to highlight the impact of size of NPs on cytocompatibilty.

Apoptosis is a molecular event characterized by changes in specific morphological features, plasma membrane blebbing, loss in membrane integrity, condensation of cytoplasm, inter-nucleosomal cleavage of DNA, etc. [29]. NPs induce apoptosis in human cells, which is generally due to long-term NP exposure to mitochondria leading to apoptotic pathway independent of caspase-8/t-Bid pathway [30]. Previously, NP-mediated toxicity has shown to induce apoptotic changes in cells [31]. A study on ZnO NP-induced toxicity reported apoptosis in human dermal fibroblasts and was further confirmed by AnnexinV staining [32]. During apoptosis, loss of membrane potential induces the translocation of membrane phospholipid phosphatidylserine (PS) from inner cellular environment to extra cellular spaces [33]. Annexin V has high affinity towards PS, thus binds to apoptotic cell's surface [15,34]. We investigated apoptotic profiles of hESC-Fib cells treated with $SiO_2$, $TiO_2$ and ZnO NPs for a desired range of concentration. As shown in Figure 6, all three NPs showed a gradual change in apoptosis along with an increase in concentration. $SiO_2$ NPs showed apoptotic response at higher concentrations as compared to $TiO_2$ and ZnO NPs. Among ZnO NP- treated cells, we observed a huge shift in apoptotic population from 10 μg/mL to 20 μg/mL group, percentage of apoptosis rises from 5% to 50% just by doubling the NP concentration. Hereby, we could conclude ZnO NPs are prone to induce drastic changes in apoptosis just by minor changes in NP concentrations.

NPs used in cosmetics, pharmaceuticals, implants, and other biomedical devices, come in contact with proliferating cells and could release into blood/serum leading to long-term toxicity [2,35]. Studies highlighted the adverse effects of NPs on cell migration properties. Destabilization of cellular migration by NPs was found due to changes which occurred in the microtubule network which lead to modification of magnitude, spatiotemporal distribution of cell movement and, ultimately, hampering wound healing capability [36]. Rise in intracellular tension due to NPs have resulted in retarding cellular migration. Live imaging of human lung carcinoma cells in presence of $SiO_2$ NPs has evidenced the changes in the dynamics of microtubule leading to decrease in cell motility [37,38]. Wound healing scratch assay showed NPs have decreased migration capacity of the cells. Among all three NPs, ZnO NPs at only 20 μg/mL have hindered 60% of cell migration. Cell migration was hindered most by ZnO > $TiO_2$ > $SiO_2$ NPs over a period of 30 h (see Figure 7). Interestingly, ZnO NPs at 10 μg/mL have induced 50% of cytotoxic effects on cell membrane integrity (confirmed by LDH assay); similarly, ZnO NPs at the same concentration have inhibited nearly 40% of wound closure. $TiO_2$ NPs at 20 μg/mL showed moderate cytotoxic effects and 50% inhibition on cellular motility.

Furthermore, NPs have been tested for genotoxic effects such as chromosomal aberration, DNA strand breaks, oxidative DNA damage, mutations, and some commonly used NPs are proven to be genotoxic at certain concentrations [39]. We performed DNA damage assay under in vitro conditions on NP-treated hESC-Fib cells. As presented in Figure 8, ZnO NPs at 20 μg/mL have shown changes in nuclear morphology and prove to be a distinct genotoxic response. In the DNA damage assay, genotoxic response is generally measured by TRITC signals and nuclear changes observed among the treated cells. Among three NPs, $SiO_2$ NPs showed negligible TRITC signals at 250 μg/mL upon 24 h treatment. However, $TiO_2$ and ZnO NPs showed strong TRITC signals. In the near future, studies will be planned to evaluate the sub-cellular changes in hESC-Fib cells by TEM imaging and for genotoxic response by real time PCR and Western blotting.

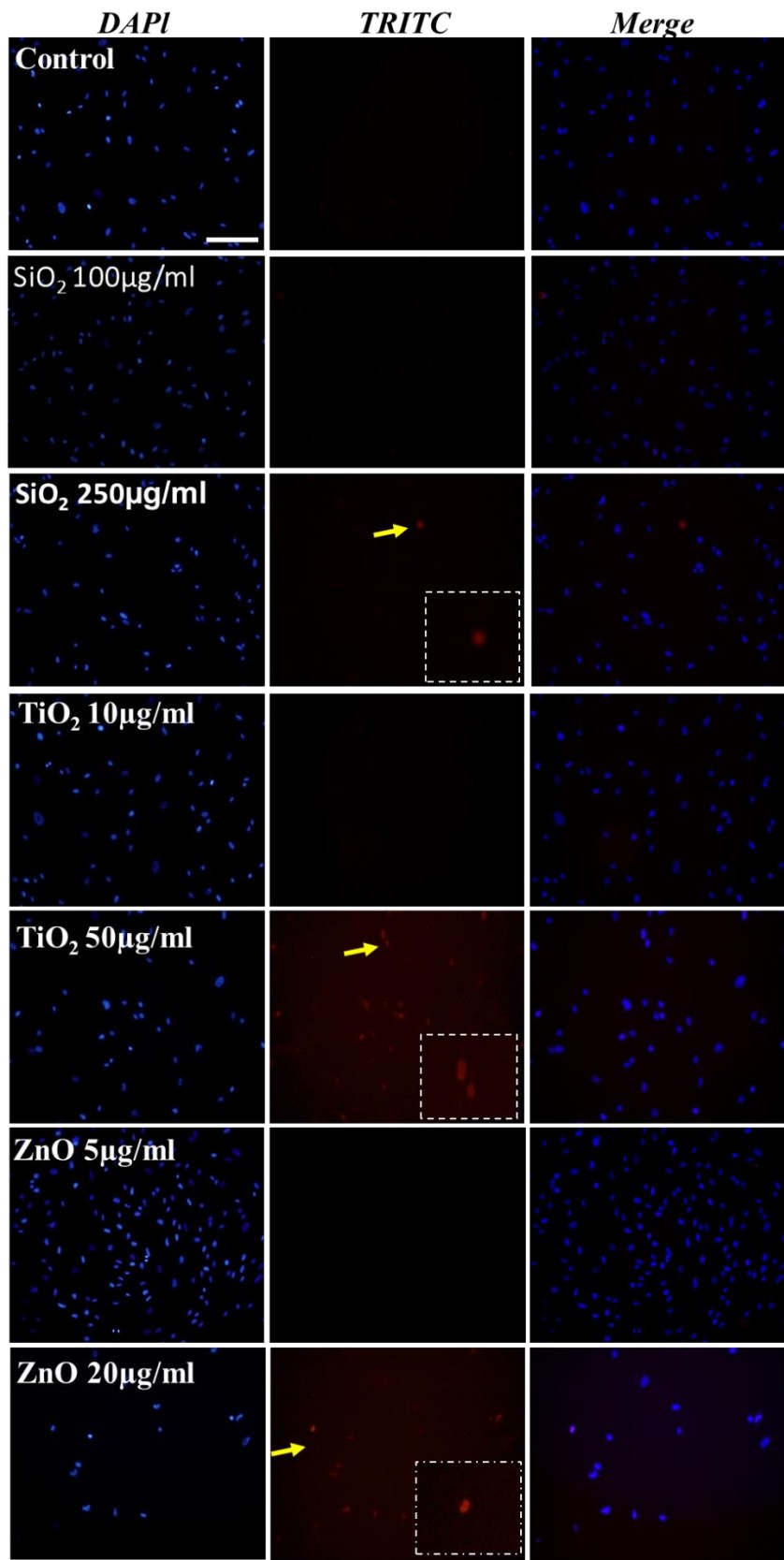

**Figure 8.** Represents the DNA damage caused by NPs on hESC-Fib cells. NP-treated hESC-Fib cells showing no genotoxic response by SiO$_2$ NPs, moderate genotoxicity from TiO$_2$ NPs and strong genotoxicity by ZnO NPs at respective IC$_{50}$ concentration. Dead cells are highlighted in yellow. Scale bar: 100 μm.

## 6. Conclusions

In this study, we have shown the concentration-dependent toxicity of ZnO, $SiO_2$ and $TiO_2$ NPs, which vary considerably, can be due to difference in size, shape, and active surface area of the nanoparticle. Wound healing scratch and MTS assay showed evidence on anti-migratory effects of NPs on hESC-fibroblasts. The membrane damage and apoptosis were mainly responsible for cell death caused by these nanoparticles. Taking into account the ability of hESC to differentiate into an unlimited number of cells that can represent how cells behave in vivo, it should be considered a promising alternative human model for predictive toxicology study of nanomaterials. This will not only lead to less use of animals in research but would also significantly reduce the drug production costs by fostering the 3Rs concept. Therefore, hESCs are a valuable source of cells which would lead to a first step towards intelligent and comprehensive in vitro testing of metal oxide nanoparticles, and in various scientific fields such as in regenerative medicine and drug screening.

**Supplementary Materials:** The following are available online at https://www.mdpi.com/2079-6412/11/1/107/s1, Figure S1: Flow cytometry graphs showing percentage of live and apoptotic cells treated with (a) $SiO_2$, (b) $TiO_2$ and (c) ZnO NPs were mentioned.

**Author Contributions:** H.K.H.: Conceptualization, Investigation, Methodology, Writing—original draft. C.A.: Formal analysis, Methodology, writing-original draft. G.S.: Conceptualization, Project administration, Writing—review & editing. C.R.K.: Conceptualization, Funding acquisition, Methodology, Project administration, Resources, Supervision. N.D.: Validation, data curation, Investigation, Methodology, Writing—review & editing. T.C.: Conceptualization, Funding acquisition, Methodology, Project administration, Resources, Supervision, Writing—review & editing. All authors have read and agreed to the published version of the manuscript.

**Funding:** This work was partially supported by grants from National University of Singapore and Singapore Ministry of Education (R221000023112, R221000026112) and National University Health System, Singapore (R221000085515, R221000085733).

**Institutional Review Board Statement:** Not applicable.

**Informed Consent Statement:** Not applicable.

**Data Availability Statement:** Data is contained within the article or supplementary material.

**Conflicts of Interest:** The authors declare no conflict of interest.

## Abbreviations

| | |
|---|---|
| hESCs | human embryonic stem cells |
| hESC-Fib | human embryonic stem cell-derived fibroblasts |
| 3Rs | Replacement: Reduction and Refinement |
| $SiO_2$ | silicon dioxide |
| $TiO_2$ | titanium dioxide |
| ZnO | zinc oxide |
| NPs | nanoparticles |
| SEM | Scanning Electron Microscope |
| FBS | fetal bovine serum |
| MTS | 3-(4:5-dimethylthiazol-2-yl)-5-(3-carboxymethoxyphenyl)-2-(4-sulfophenyl)-2H-tetrazolium |
| LDH | lactate dehydrogenase |
| OECD | Organization for Economic Co-operation and Development |
| ROS | reactive oxygen species |
| DMEM | Dulbecco's Modified Eagle Medium |
| NOAEC | no observable adverse effect concentration |
| $IC_{50}$ | 50% inhibitory concentration |
| TLC | total lethal concentration |
| FACS | fluorescence-assisted cell sorting |

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
