# Peer review of "Cytotoxicity and Genotoxicity of Metal Oxide Nanoparticles in Human Pluripotent Stem Cell-Derived Fibroblasts"

_coatings, doi:10.3390/coatings11010107_

Round 1

Reviewer 1 Report

The authors address a highly interesting and needed study concerning to the toxicity of three types of nanoparticles: ZnO, TiO2 and SiO2. Despite the interest of the subject, the manuscript lacks several aspects and should be deeply revised before its consideration for publication. In addition, the subject does not seem appropriate for a journal named Coatings because the particles under study are naked.

Major points.

1. In general, the text shows irrelevant or too basic information for a research publication while relevant data is missing.

2. The particles analyzed have been purchased and have been extensively studied and characterized previously, thus the study lacks in originality as far as the nanoparticles is concerned.

3. Lines 173 to 179. Concentrations are quite confusing. In line 173, it is stated that concentrations up to 50 μg/mL can be used without damaging the cells. However, in line175, 50 μg/mL is considered as toxic.

4. There is also a lack of appropriate controls all over the text.

5. Language should be carefully revised. Some words lead to a text misunderstanding.

Minor points.

6. Figure 5 is not sharp enough.

7. Figure legend 6 has obvious information such as the meaning of the axis.

8. There are references related to the subject more recent that the referred by the authors. Apart from one of 2019, the rest have been published before 2016.

Author Response

Dear Editor & Reviewers,

We would like to thank you for the suggestions and comments. We have managed to address all the suggested comments. Kindly, find the responses to the comments below.

Reviewer 1

The authors address a highly interesting and needed study concerning to the toxicity of three types of nanoparticles: ZnO, TiO2 and SiO2. Despite the interest of the subject, the manuscript lacks several aspects and should be deeply revised before its consideration for publication. In addition, the subject does not seem appropriate for a journal named Coatings because the particles under study are naked.

Major points.

In general, the text shows irrelevant or too basic information for a research publication while relevant data is missing.

1) The particles analysed have been purchased and have been extensively studied and characterized previously, thus the study lacks in originality as far as the nanoparticles is concerned.

Response: Thank you for the critical analysis of our manuscript. Since there is a great need for novel in vitro methods to predict human developmental toxicity to comply with the 3R (Replacement, Reduction and Refinement), principles and to improve human safety.

As mentioned in introduction “Due to high volumes of testing and limited availability of primary human cells, cell lines of animal-origin, immortalized and/or cancer-derived sources are commonly used for high-throughput toxicology screening studies. Since, the cellular immortality leads to epigenetic changes, tumorigenesis, chromosomal and genetic aberrations, the physiological relevance, and reliability of cancer-derived and immortalised cell lines is limited. Further, though primary human cells are a good platform to rely on, procuring large amounts of primary cells typically required for high-throughput assays is challenging. The pluripotent stem cells such as human embryonic stem cells (hESCs) due to their long-term self-renewal and pluripotent ability to derive differentiated lineages, are potential source for obtaining unlimited amounts of somatic cells and are more significant in reflecting human physiological milieu.

This present study is unique & novel since there are limited reports on toxicity testing of the commercially available nanoparticles on hESCs derived fibroblast lineage.

2) Lines 173 to 179. Concentrations are quite confusing. In line 173, it is stated that concentrations up to 50 μg/mL can be used without damaging the cells. However, in line 175, 50 μg/mL is considered as toxic.

Response (Line 173 to 176): Thank you for the comment. The 50 μg/mL in line 173 is for SiO2 and in line 175 is for ZnO. The statement is modified to clarity.

“The SiO2, TiO2 and ZnO NP treated cells displayed morphology close to that of untreated fibroblasts at a concentration of 50, 10, and 5 μg/ml. However, detrimental effect on cell morphology and even cell death was observed with higher NP concentrations (i.e.,1250, 250 and 50 μg/ml SiO2, TiO2 and ZnO NPs respectively).”

3) There is also a lack of appropriate controls all over the text.

Response: Thank you for the comment. Cells free of nanoparticles were used as control cells (100 % viability).

4) Language should be carefully revised. Some words lead to a text misunderstanding.

Response: Thank you.

Minor points.

5) Figure 5 is not sharp enough:

Response (Line 194): Thank you for pointing out the quality of the figures. It has been corrected in the revise manuscript.

6) Figure legend 6 has obvious information such as the meaning of the axis.

Response (Line 209): Thank you for the comment. The caption has been modified in the revise manuscript

“Figure 6:  Effect of nanoparticles on the apoptosis of hESC-derived fibroblast exposed to different concentrations in culture after 24 hours. “

7) There are references related to the subject more recent that the referred by the authors. Apart from one of 2019, the rest have been published before 2016.

Response: Thank you for the comment. The references have been modified to add the most recent literature.

Reviewer 2 Report

The manuscript by Handral et al. aimed to investigate the cytotoxicity and genotoxicity of metal oxide nanoparticles by using the human pluripotent stem cell-derived fibroblasts as in vitro model. This study is of interest; however, there are several shortcomings needing to be experimentally addressed for increasing the impact of this study.

  1. The statistical analysis used in this study is incorrect; a biostatistician is highly recommended.
  2. In figure 5, the symbol for the results of statistical analysis should be labeled in graph.
  3. In figure 6, the images of flow cytometry and apoptotic cells (by TUNNEL assay) are required for this study.
  4. In figure 7, wound healing scratch assay could measure the capability of both cell proliferation and migration, not only for cell proliferation; please correct it. In addition, the images for wound healing scratch assay are required for this study.
  5. In figure 8, the quantitative data for DNA damage are required. In addition, it will be nice to provide the higher magnification of images in this figure.

Author Response

Dear Reviewer,

We would like to thank you for the suggestions and comments. We have managed to address all the suggested comments. Kindly, find the responses to the comments below.

1.The statistical analysis used in this study is incorrect; a biostatistician is highly recommended.

Response: Thank you for the comment. The statistical analysis was performed using One-way Anova and Tukey test (α = 0.05). We have included information in the manuscript (Line 142 & 143), and in the respective figure legend as well (Line 210).

2. In figure 5, the symbol for the results of statistical analysis should be labeled in graph.

Response: Thank you. We have updated the Figure 5 with the labels of statistical analysis. Please refer to updated Figure 5.

3.In figure 6, the images of flow cytometry and apoptotic cells (by TUNNEL assay) are required for this study.

Response: Thank you for the comment. Annexin V staining is one of the most common methods used for detecting apoptotic cells showing the quantitative data of apoptotic cells. Annexin V assay also provides a descriptive information about cells on percentage of viable cells, early and late apoptotic and even necrotic cells. And this assay is also considered as an alternative assay to TUNEL assay. Flow cytometry results were presented in graphical representation for a better comparative analysis.

4. In figure 7, wound healing scratch assay could measure the capability of both cell proliferation and migration, not only for cell proliferation; please correct it. In addition, the images for wound healing scratch assay are required for this study.

Response: Thank you for the insightful comment. We have corrected the information on the wound healing scratch assay and provided the images of the scratch assay.

5. In figure 8, the quantitative data for DNA damage are required. In addition, it will be nice to provide the higher magnification of images in this figure.

Response: Thank you for the comment. Magnified section of damaged cells has been included as an inset in the existing figures, which can provide better clarity on the results.

Reviewer 3 Report

The authors investigated the toxicity of three kinds of nanoparticles (ZnO, TiO2, and SiO2) by using of pluripotent human embryonic stem cell (hESCs) derived fibroblasts. Cytotoxicty assays, flow cytometry, and DNA damage assay were used to understand the toxicity profiles of the nanoparticles. The idea and motivation are reasonable, and the characterizations are well performed. The paper will need to undergo minor revision before it can be published in Coatings.

  1. The authors should add the discussion about why smaller size of SiO2 was chosen to compared with ZnO and TiO2.
  2. The authors should add the discussion about why rod shape of ZnO was chosen to compared with spherical shape of SiO2 and TiO2.
  3. Higher resolution of SEM images should be provided.
  4. The image resolution of figure 6 should be adjusted.

Author Response

Dear Reviewer,

We would like to thank you for the suggestions and comments. We have managed to address all the suggested comments. Kindly, find the responses to the comments below.

1) The authors should add the discussion about why smaller size of SiO2 was chosen to compare with ZnO and TiO2.

Response (Line 261 & 262): We have provide a brief explanation on why small sized SiO2 nanoparticles were used for the study.

“Small sized SiO2 nanoparticles are widely used compared to bigger size and have been proved more toxic than bigger SiO2 nanoparticles.”

2) The authors should add the discussion about why rod shape of ZnO was chosen to compare with spherical shape of SiO2 and TiO2.

Response (Line 256 to 260): Thank you for insightful comment. Based on the literature, rod-shaped ZnO NPs show higher toxicity effects. Below text is added in the revised manuscript to address why nanorod ZnO was used.

 “The rod-shape ZnO nanoparticles have shown to have higher toxicity compare to the corresponding spherical ones. Thus, nano-rod ZnO particle was to assess the toxicity response on hESC derived fibroblast compared to spherical shaped SiO2 or TiO2 to induce significant toxicity”

3) Higher resolution of SEM images should be provided.

Response: Thank you for the comment. As TEM can provide a higher resolution and is better suited for particle size analysis in the nano particle we have provide the high resolution TEM images (Figure 2) of the nanoparticles.

4) The image resolution of figure 6 should be adjusted.

Response (Line 209): Thank you for pointing out the quality of the figures. It has been corrected in the revise manuscript.

Round 2

Reviewer 1 Report

The authors have addressed most of the comments raised and the manuscript has been substantially improved, but there are still some unclear points.

1. How is the concentration of the particles calculated, from the purchased sample?

2. In the legend of figure 2 it is said that "The agglomeration of SiO2 and TiO2 NPs was also represented". In general, the figures show a high polydisperse population, therefore the samples are agglomerated.

3. Letters in figure 5 are difficult to read

4. References should be carefully revised. Most of them are uncomplete (lack the page numbers) 9, 11, 14, 23, 27…... N. 30 has no authors, in some there is only one author (5, 10, 17…) while in others, several authors are mentioned (6, 23…)

Author Response

Dear Editor & Reviewers,

We would like to thank you for the suggestions and comments. We have managed to address all the suggested comments. Kindly, find the responses to the comments below.

Reviewer 1

The authors have addressed most of the comments raised and the manuscript has been substantially improved, but there are still some unclear points.

1. How is the concentration of the particles calculated, from the purchased sample?

Response: The detailed description of concentration preparation is described in the revised manuscript. (Line 61 to 65)

“A stock dispersion of as received nanoparticles was weighed on an analytical mass balance, suspended in deionized water at a concentration of 50 mg/ml, and then probe sonicated for 2 min at 35–40 W to assist in mixing and forming a homogeneous dispersion. For further use the stock concentration was diluted to prepare working concentrations in DMEM-high glucose media with 10% FBS and 1% pen/strep”.

2. In the legend of figure 2 it is said that "The agglomeration of SiO2 and TiO2 NPs was also represented". In general, the figures show a high polydisperse population, therefore the samples are agglomerated.

Response: Thank you for this comment. Legend of Figure 2 has been modified.

3. Letters in figure 5 are difficult to read

Response: Thank you for this comment. We have modified the figure.

4. References should be carefully revised. Most of them are uncomplete (lack the page numbers) 9, 11, 14, 23, 27…... N. 30 has no authors, in some there is only one author (5, 10, 17…) while in others, several authors are mentioned (6, 23…)

Response: Thank you for this comment. After careful revision, we have updated the reference with the correct information.

Reviewer 2 Report

Authors have addressed most of the comments; however, authors did not provide the representative images of flow cytometry in figure 6. 

Author Response

Cover letter for revision of manuscript

Dear Editor & Reviewers,

We would like to thank you for the suggestions and comments. We have managed to address all the suggested comments. Kindly, find the responses to the comments below.

Reviewer 2

Authors have addressed most of the comments; however, authors did not provide the representative images of flow cytometry in figure 6. 

Response: Thank you for this comment. We have included the flow cytometry micrographs of figure 6 in the supplementary information.
